# Sleep Quality and Insomnia Severity among Italian University Students: A Latent Profile Analysis

**DOI:** 10.3390/jcm11144069

**Published:** 2022-07-14

**Authors:** Matteo Carpi, Daniel Ruivo Marques, Alberto Milanese, Annarita Vestri

**Affiliations:** 1Department of Psychology, Sapienza University of Rome, 00185 Rome, Italy; 2Department of Education and Psychology, Campus Universitário de Santiago, University of Aveiro, 3810-193 Aveiro, Portugal; drmarques@ua.pt; 3CINEICC—Center for Research in Neuropsychology and Cognitive Behavioral Intervention, Faculty of Psychology and Educational Sciences, University of Coimbra, 3000-115 Coimbra, Portugal; 4Department of Biomedical and Clinical Sciences “L. Sacco”, University of Milan, 20122 Milan, Italy; alberto.milanese@unimi.it; 5Department of Public Health and Infectious Diseases, Sapienza University of Rome, 00185 Rome, Italy; annarita.vestri@uniroma1.it

**Keywords:** insomnia, sleep quality, university students, latent profile analysis, symptom profiles

## Abstract

Insomnia is a widespread sleep disorder associated with physical and mental health conditions. Although the heterogeneity of insomnia presentations has been acknowledged, research investigating clinically meaningful insomnia subtypes is still ongoing. This study aimed at exploring insomnia subtypes according to widely-used measures of symptoms severity and sleep quality among Italian university students using a latent profile analysis. Data were collected from 490 students reporting relevant insomnia symptoms through an online cross-sectional survey comprising the Insomnia Severity Index, the Pittsburgh Sleep Quality Index, the 21-item Depression Anxiety Stress Scale, and the Short Form-12. Latent profile analysis identified five insomnia subtypes. The severe insomnia (8.8%) group showed the highest insomnia severity, with diverse complaints concerning sleep quality and daytime functioning. Moderate insomnia with sleep duration complaints (8.4%) and moderate insomnia with medication use (15.9%) subgroups were characterized by middle range insomnia severity, with problems of sleep continuity and sleep medication use, respectively. Subthreshold insomnia with sleep latency complaints (20.4%) and subthreshold insomnia (46.5%) groups showed attenuated insomnia symptoms. Higher psychological complaints and worse quality of life were associated with greater sleep complaints. Overall, these findings highlight the relevance of sleep quality domains in identifying insomnia subtypes and might help optimize insomnia treatments.

## 1. Introduction

Insomnia is a widespread sleep disorder, with an average prevalence rate of 10% [1,2,3] and even higher prevalence estimates reported after the COVID-19 outbreak [4]. Given its diffusion and its well-documented associations with adverse outcomes for both mental and physical health [5,6,7,8,9,10], insomnia is presently considered a relevant public health issue [2,11]. Interestingly, young adults [12] and university students in particular [13] report insomnia complaints more frequently in comparison with the general population, and according to several studies, the COVID-19 pandemic had a significant impact on this population’s sleep quality [14,15]. Given that early diagnosis and treatment of sleep difficulties could prevent adverse outcomes in the long term [6], collecting data among university students is critical for both research and clinical practice.

According to current diagnostic systems, insomnia disorder is characterized by a predominant complaint of dissatisfaction with sleep quality or quantity in association with difficulty initiating sleep, difficulty maintaining sleep, or early-morning awakenings with inability to return to sleep occurring despite adequate opportunity. For the diagnosis, these difficulties should cause clinically significant distress and impairment in daytime functioning and should be present for at least three nights per week [16,17].

Although several insomnia subtypes based on diagnostic features and clinical presentation (e.g., sleep-onset insomnia and sleep maintenance insomnia) have been proposed, their reliability is not apparent, and no clinically significant differences were found between them [18]. On the other hand, two insomnia phenotypes have been identified considering sleep duration: one showing both cognitive-emotional and physiological hyperarousal with objective short sleep duration assessed by polysomnography and the other characterized by cognitive-emotional arousal, normal sleep duration with sleep misperception, and anxiety/rumination. Worse health outcomes in the long term and a non-remitting course have been consistently observed in association with the objective short-sleep-duration phenotype [7,19]. However, the identification of insomnia phenotypes is still ongoing and represents a significant trend in contemporary insomnia research [20,21,22].

In this context, the relevance of subjective perceptions for the diagnosis of insomnia and the wide availability of dedicated self-report instruments [23] encourage taking into account a broad set of factors in evaluating insomnia phenotypes. Both objective and subjective sleep and non-sleep variables might be considered, including symptoms, sleep macrostructure and microstructure, environmental factors, personality traits, behaviors, attitudes, and genetics. Such a data-driven approach has already provided promising results for other disorders such as major depression [24] and executive function-related difficulties in children [25].

Indeed, beyond top-down-defined phenotypes, several studies sought to investigate insomnia subtypes with bottom-up, data-driven methods, such as cluster analysis and latent class analysis, to explore the role of diverse factors in determining insomnia presentation and severity [26,27,28,29,30]. While cluster analysis on objective sleep parameters confirmed the relevant distinction between insomnia with short sleep duration and insomnia with normal sleep duration [26], the addition of subjective measures allowed to ascertain more subtle aspects contributing to identification of clinically meaningful insomnia subtypes (e.g., daytime sleepiness, sleep fragmentation, fatigue, daytime impairment, psychological distress) [27,28,29]. Moreover, in line with recent indications about insomnia-relevant individual characteristics [20], recent results obtained with latent class analysis considering both sleep and non-sleep variables (including negative affect and personality traits) identified longitudinally stable subtypes showing significant differences in outcome variables such as response to treatment and risk of depression [30].

As a whole, these findings highlight the relevance of subjective variables in identifying clinically meaningful insomnia profiles and the already well-documented importance of the relationship between insomnia severity and subjective complaints of daytime impairment and distress. However, the reviewed studies did not thoroughly explore subjective sleep quality as measured by the Pittsburgh Sleep Quality Index (PSQI) [31] to characterize insomnia. Despite the definition of sleep quality being still debated [32,33], the PSQI is a widely used and easily administered instrument that measures a set of sleep-related characteristics pertaining to perceived sleep quality, sleep continuity, and sleep-related complaints. Given its practicality and suitability for routine assessment, subtypes identified by the PSQI components scores might be particularly informative and helpful for clinical practice in community settings. For example, Chen, Hsu, and Chou [34] conducted a latent class analysis with the seven PSQI sub scores on a sample of 1011 older adults with poor sleep quality (i.e., PSQI total score > 5) and found three groups that were labeled, respectively, “high insomnia”, “mild insomnia”, and “high hypnotics”. High-insomnia and high-hypnotics groups showed similar mental and physical impairments in comparison with the mild-insomnia group and with a community sample of older adults with fair sleep quality. Moreover, high-insomnia participants also showed worse insomnia severity and worse sleep quality (in terms of sleep satisfaction, perceived sleep continuity, and daytime dysfunction). Nevertheless, the authors relied only on the PSQI components and did not use a dedicated measure of insomnia symptoms. Considering the limited efficacy of the PSQI in screening clinical insomnia [35], relevant insomnia features might have been disregarded in the subgroups they identified.

Starting from these results, this study aims at exploring insomnia profiles considering both sleep quality components as measured by the PSQI and insomnia severity in a sample of Italian university students reporting relevant insomnia symptoms and to investigate the differences between these profiles with respect to psychological functioning (i.e., symptoms of anxiety, depression, and stress) and health-related quality of life.

## 2. Materials and Methods

### 2.1. Participants and Procedure

This study is part of a research project aimed at investigating psychological well-being, sleep quality, and health habits of students enrolled at Sapienza University of Rome. Participants were 490 university students (mean age: 23.4 ± 2.4 years, 432 women and 58 men) attending undergraduate and postgraduate taught courses selected from a larger database (*n* = 1288). After excluding all subjects who provided invalid responses (*n* = 68), participants younger than 35 years old and reporting relevant symptoms of insomnia (Insomnia Severity Index [36] score ≥ 10 as recommended by Morin and colleagues [37] for community samples) were selected in line with the purpose of this study. No further exclusion criteria were applied.

The study design was cross-sectional, and the research procedure was reported in detail previously [38]. Data were collected via an anonymous online survey delivered to students from Sapienza University of Rome through the Google Forms platform from March 2021 to June 2021. This procedure allows for rapidly collecting data from large samples using adapted versions of paper-and-pencil questionnaires and instruments and has been widely used in psychology and sleep research [4,8,29,39]. Participants took part in the study on a voluntary basis and provided online informed consent. The study procedures were approved by the competent Ethics Committee at Sapienza University of Rome (protocol number 0308/2021).

### 2.2. Measures

Socio-demographic data and information about academic career, health status and health-related habits (e.g., height, weight, alcohol and tobacco consumption, physical exercise) were obtained through dedicated questions, whereas sleep, psychological symptoms, and health-related quality of life were investigated with the Italian versions of the Insomnia Severity Index (ISI) [36], the Pittsburgh Sleep Quality Index (PSQI) [31], the 21-item Depression Anxiety Stress Scale (DASS-21) [40], and the Short Form-12 questionnaire (SF-12) [41]. The reliability of the scales in the original sample (*n* = 1288) was evaluated with respect to conventional criteria [42] through Cronbach’s alpha (*α*) [43] and McDonald’s omega (*ω*) [44] coefficients.

#### 2.2.1. Insomnia Severity Index

Insomnia symptoms were assessed with the Insomnia Severity Index (ISI) [36], a seven-item self-report questionnaire measuring subjective sleep difficulties. Items are rated on a five-point response scale from 0 to 4 with higher scores corresponding to greater symptom severity, and their sum yields a global score ranging from 0 to 28. The first three items measure insomnia severity (difficulties in initiating and maintaining sleep and waking up too early), and the last four items assess sleep satisfaction, noticeability of the sleep problem to others, worry about the sleep problem, and sleep problem’s interference with daily functioning. Four severity categories have been identified for the total score: no insomnia (score range 0–7), subthreshold insomnia (score range 8–14), moderate insomnia (score range 15–21), and severe insomnia (score range 22–28), and the original version of the instrument showed acceptable reliability, convergent validity with other subjective and objective sleep measures, and sensitivity to change after treatment. According to Morin et al. [37], a cut-off of 10 maximizes sensitivity and specificity and is optimal for detecting insomnia cases in community samples, and thus a total score ≥ 10 was used in this study to select subjects reporting relevant insomnia symptoms.

The Italian version of the ISI demonstrated acceptable internal consistency (*α* = 0.75) and convergent validity with sleep diaries measures, and the instrument showed good reliability in this study (*α* = 0.84, *ω* = 0.84).

#### 2.2.2. Pittsburgh Sleep Quality Index

Sleep quality was assessed with the Pittsburgh Sleep Quality Index (PSQI) [31]. This self-report instrument comprises 19 items with different response formats (5-point Likert scales or open-ended) that investigate perceived sleep quality, sleep time habits, sleep problems, and sleep-related disturbances and measures seven dimensions (subjective sleep quality, sleep latency, sleep duration, habitual sleep efficiency, sleep disturbances, use of sleep medications, and daytime dysfunction) with aggregated scores ranging from 0 to 3 for each. The sum of the dimensions’ scores provides a total score. Higher scores indicate worse sleep quality, and a cut-off of 5 was identified for poor sleep quality. In order to investigate specific aspects of sleep quality, the seven dimensions’ scores were used in this study, and information about sleep variables (e.g., sleep latency and sleep duration) was derived from the PSQI items.

The Italian version of the PSQI showed good reliability (*α* = 0.83) and is able to discriminate between patients with sleep disorders and healthy controls [45]. In this study, reliability for the seven dimensions’ scores was acceptable (*α* = 0.70, *ω* = 71).

#### 2.2.3. Depression Anxiety Stress Scale-21

Symptoms of anxiety, depression, and stress were evaluated with the 21-item version of the Depression Anxiety Stress Scale (DASS-21) [40]. Items refer to the previous week and are rated on a four-point Likert scale from 0 to 3. Each one of the scale dimensions (namely anxiety, depression, and stress) is measured by seven items, and dimensions’ scores are obtained by summing the items’ responses.

The Italian DASS-21 showed satisfactory reliability (*α* of 0.74, 0.82, and 0.85, respectively, for the anxiety, depression, and stress subscales and *α* = 0.90 for the total score) and convergent validity with other measures of distress and psychopathology [46]. In this study, the instrument demonstrated good reliability (*α* = 0.85 and *ω* = 0.85 for anxiety, *α* = 0.90 and *ω* = 0.90 for depression, *α* = 0.87 and *ω* = 0.87 for stress, and *α* = 0.94 and *ω* = 0.94 for the total score).

#### 2.2.4. Short Form-12

The Short Form-12 (SF-12) [41] was used to assess health-related quality of life. Items are rated on different response formats (three- to six-point scales and yes/no) and measures eight dimensions (physical activity, role and physical health, role and emotional state, mental health, physical pain, general health, vitality, social activities) whose scores are aggregated into two indexes, namely the physical component summary (PCS) and the mental component summary (MCS), reflecting, respectively, physical and mental aspects of health-related quality of life. Scores for the two components are standardized (with a mean of 50 and a standard deviation of 10) according to the scoring procedure described by Ware et al. [47], and higher scores correspond to better quality of life.

The Italian adaptation of the questionnaire [48] was used in this study, and good reliability was found for the total score (*α* = 0.81, *ω* = 0.82).

### 2.3. Statistical Analyses

Descriptive statistics were computed to explore sample characteristics. Means and standard deviations or medians and interquartile ranges were obtained for continuous variables, whereas categorical variables were summarized by counts and percentages. Latent profile analysis (LPA) was used to classify the participants into homogeneous subgroups with respect to sleep characteristics and complaints. LPA is a statistical clustering approach that identifies groups of individuals (i.e., classes or profiles) based on their pattern of scores in a set of indicator variables assigning each individual to a profile with a certain degree of probability [49]. PSQI components’ scores and the ISI total score were considered as indicator variables in this study. *Z*-scores (mean = 0, standard deviation = 1) were obtained for each variable to simplify results interpretation, and models with 2 to 6 classes were estimated consistently with previous results reported in the field of sleep research [27,28,29,34,50,51]. Equal variances were assumed for all the indicators, and covariances were fixed to 0. To identify the optimal number of profiles, conceptual coherence and clinical meaningfulness were evaluated, and several fit indices were used, namely the Akaike information criteria (AIC), the Bayesian information criteria (BIC), the sample size-adjusted Bayesian information criteria (SABIC), the bootstrapped likelihood ratio test (BLRT), and entropy (i.e., the ability of the model to identify well-separated profiles [52]). Lower AIC, BIC, and SABIC values; significant BLRT *p*-values; and higher entropy values were considered as signs of a better fit [49,53].

Chi-square tests and one-way analyses of variances (ANOVA) were used to explore the significance of the differences in sleep variables (PSQI total and components scores, ISI total score), psychological symptoms (DASS-21 anxiety, depression, and stress scores), health-related quality of life (SF-12 PCS and MCS scores), and health habits (physical exercise, smoking habit, and alcohol consumption) between participants in the profiles identified with LPA. Before conducting the ANOVAs, residuals’ distributions for the dependent variables were examined, and variables showing relevant deviations from normality were log-transformed. Post hoc tests were conducted using Hochberg’s GT2 test to control for different sample sizes [54], and eta-squared (*η^2^*) effect sizes were computed.

For all the analyses performed, *p*-values below 0.05 were considered statistically significant. Descriptive statistical analyses, ANOVAs, and chi-square tests were conducted with IBM SPSS software (version 25.0, IBM Corp., Armonk, NY, USA), and LPA was performed with R [55] using the tidyLPA package [56].

## 3. Results

### 3.1. Sample Characteristics

Participants’ demographic and health-related characteristics are reported in Table 1. As already reported, the large majority of the sample was women (88.2% women, 11.8% men), and participants were enrolled in undergraduate (83.9%) and postgraduate (16.1%) courses.

Mean scores of the questionnaires measuring sleep habits and complaints, psychological symptoms, and health-related quality of life are shown in Table 2 (cf. total sample). According to the ISI total scores, 67.1% (95% CI: 62.9–71.3%; *n* = 329) of the sample reported subthreshold insomnia symptoms (score range 10–14), 30.2% (95% CI: 26.1–34.3%; *n* = 148) moderate insomnia symptoms (score range 15–21), and 2.7% (95% CI: 1.3–4.3%; *n* = 13) severe insomnia symptoms (score > 21).

### 3.2. Insomnia Profiles Characterization

Fit indices for the evaluated LPA solutions are reported in Table 3. The five-classes model showed a better fit, with lower values of AIC, BIC, and SABIC and higher entropy in comparison with the other models. In addition, a non-significant BLRT value was found for the six-classes model, further confirming the five-classes model’s adequacy. Given these results and also considering the relative size of the five identified classes (all classes comprised more than 8% of the sample) and their conceptual relevance, the five-profiles solution was selected as the optimal model.

Classes centroids (i.e., mean *z*-scores of the indicator variables) are represented graphically in Figure 1. On the basis of the sleep-related indicators’ configurations in each profile, classes were named as follows: (1) *severe insomnia* (SI; 8.8% of the sample), comprising students reporting high overall insomnia severity, above-average sleep latency and notably above-average complaints related to sleep quality, sleep disturbances, and daytime dysfunction; (2) *moderate insomnia with medication use* (MI-MU; 15.9% of the sample), including students with above-average insomnia severity and slightly above-average scores in all the PSQI components except for their high use of sleep medication; (3) *subthreshold insomnia* (SubI; 20.4% of the sample), composed by students with relatively low insomnia severity and below-average sleep complaints; (4) *subthreshold insomnia with sleep latency complaints* (SubI-SL; 46.5% sample), comprising students with comparatively low levels of insomnia symptoms showing slightly above-average sleep latency and below-average complaints in the other PSQI components; (5) *moderate insomnia with sleep duration complaints* (MI-SD; 8.4%), made up of those participants who reported above-average insomnia severity with above-average sleep latency and sleep quality complaints and considerably above-average problems with sleep duration and sleep efficiency.

Between-classes mean differences in sleep variables are reported in the first part of Table 2. All the *F* statistics computed by the ANOVAs were statistically significant, with *p*-values below 0.001. According to post hoc analyses, the severe insomnia class (SI) showed significantly higher overall insomnia severity than the moderate insomnia classes and the subthreshold insomnia classes, whereas the moderate insomnia classes had worse overall sleep quality (i.e., higher PSQI total score) than the severe insomnia class and the two subthreshold classes. In the aggregate, similar insomnia severity and overall sleep quality were observed within profiles with moderate sleep complaints (MI-SM and MI-SD) and profiles with milder sleep complaints (SubI and SubI-SL, showing lower ISI and PSQI total scores).

PSQI components’ score patterns were consistent with profiles characterization. Notably, the SI class showed worse perceived sleep quality, sleep disturbances, and daytime dysfunction in comparison with the other classes, the MI-MU class showed higher sleep medication use, and the MI-SD class had higher mean scores in sleep duration and habitual sleep efficiency (i.e., shorter sleep and reduced sleep efficiency).

### 3.3. Differences between Insomnia Profiles in Health Habits, Psychological Complaints, and Health-Related Quality of Life

A significant association was found between class membership and sex (*χ*^2^ = 11.8, *p* < 0.05). Whereas sex distribution in the SI, SubI, and SubI-SL classes was akin to that reported for the total sample, a higher proportion of women was observed in the MI-MU (96.2% women vs. 3.8% men) and the MI-SD (97.6% women vs. 2.4% men) classes. On the other hand, no associations were found between class and physical exercise (*χ*^2^ = 2.6, *p* = 0.63), tobacco use (*χ*^2^ = 8.4, *p* = 0.08), and excessive alcohol consumption (*χ*^2^ = 7.2, *p* = 0.13), and no differences in age (*F* = 0.8, *p* = 0.56, *η*^2^ = 0.01) and body mass index (*F* = 0.4, *p* = 0.83, *η*^2^ = 0.01) were found between classes. Demographic and health-related characteristics for each profile are reported in detail in Table A1 (Appendix A).

The second part of Table 2 shows mean differences between classes in DASS-21 and SF-12 scores. Significant differences between classes were found for all the variables (*p* < 0.05 for the MCS, *p* < 0.001 for the other variables). With respect to the DASS-21 scores, post hoc analyses showed that participants classified in the SI class had higher anxiety than those in the other classes, whereas participants in the MI-MU class reported higher anxiety only in comparison with those in the subthreshold symptoms classes SubI and SubI-SL; the SI class also showed a higher mean depression score than the MI-MU, the SubI, and the SubI-SL classes and a higher mean stress score than the SubI, the SubI-SL, and the MI-SD classes; and again, the MI-MU class had a significantly higher mean stress score than the SubI and the SubI-SL classes.

Regarding the SF-12 scores, participants in the SI class reported lower scores in both physical (PCS) and mental (MCS) health-related quality of life in comparison with participants in the SubI and the SubI-SL classes.

## 4. Discussion

This study sought to explore insomnia subtypes in a sample of university students reporting clinically relevant insomnia complaints considering self-report sleep quality and insomnia severity and using a data-driven approach. Consistently with the adopted inclusion criteria (ISI score ≥ 10), the sample means of self-reported total sleep time and sleep-onset latency were, respectively, below seven hours and above 30 min and can be considered indicative of poor sleep according to previously reported data [57,58].

The results of latent profile analysis confirmed a significant heterogeneity in insomnia presentations according to the dimensions considered (i.e., the ISI and the PSQI components) and identified five insomnia subtypes showing distinguishable characteristics with three degrees of symptoms severity (severe, moderate, and subthreshold). The *severe insomnia* (SI) subtype was characterized by high overall insomnia severity with various complaints, showing, in particular, a poor perceived sleep quality with significant and diverse symptoms of sleep disturbances and relevant daytime impairment. This group also showed higher levels of psychological problems (i.e., anxiety, depression, and distress) and worse physical and mental health-related quality of life in comparison with the others. As a whole, this pattern of symptoms and complaints might yield a significant impact on well-being and seemingly fulfill the criteria for insomnia disorder [2], showing a major overlap with previously identified insomnia subtypes characterized by high distress and psychological comorbidities [27,29].

In the middle range of insomnia severity, the *moderate insomnia with medication use* (MI-MU) and the *moderate insomnia with sleep duration complaints* (MI-SD) subgroups showed different characteristics. The latter (MI-SD) was characterized in particular by complaints concerning sleep duration and continuity (peak scores in the PSQI components of sleep duration and sleep efficiency) and sleep quality. On the other hand, the MI-MU profile exhibited a frequent use of sleep medication (a distinctive trademark as compared to other subgroups) with attenuated sleep complaints but an analogous level of daytime dysfunction in comparison with the MI-SD profile. However, both the moderate insomnia profiles obtained comparable DASS-21 and SF-12 scores. Thus, despite different sleep complaints, the two subgroups appear similar in terms of functional impairment. This result is somewhat in line with that reported by Chen and colleagues [34] with respect to subjects reporting high-frequency sleep medication use in their study and also with previous research showing that pharmacological sleep treatments are indeed effective on nocturnal insomnia symptoms but may yield adverse effects with a possible impact on daytime functioning [59,60,61].

Finally, the subthreshold symptoms subgroups showed mild sleep complaints with lower distress and better health-related quality of life in comparison with the other profiles. In particular, the *subthreshold insomnia with sleep latency complaints* (SubI-SL) profile (comprising nearly half of the sample) was characterized by relevant difficulty initiating sleep with mild overall sleep complaints, whereas individuals in the *subthreshold insomnia* (SubI) group did not show significant sleep complaints but still reported daytime dysfunction scores similar to those of the other subgroups. This last group might be composed of individuals who were not facing relevant insomnia symptoms (false positives) or may otherwise comprise those reporting sleep complaints not primarily characterized by insomnia symptoms that have not been explored in this study (e.g., excessive daytime sleepiness).

On the whole, the identified insomnia subtypes recall symptoms clusters previously reported on the basis of objective and subjective sleep measures (in particular, the three clusters described by Crawford et al.) [27,28,29,51] and expand them, highlighting the relevance of sleep quality components such as sleep medication use and daytime dysfunction. Additionally, in line with the previous literature [6,8,9], worse mental health and health-related quality of life were observed in the subtypes characterized by a higher insomnia severity in a seemingly gradient-like fashion. No differences between subgroups were found in health-relevant behaviors (physical exercise, tobacco use, and alcohol consumption) and body mass index, whereas higher proportions of women were observed in the two moderate symptoms profiles (MI-MU and MI-SD). Although a higher insomnia prevalence among women has been well-documented [62], to our knowledge, no associations between female sex and the specific characteristics of these profiles (i.e., frequent sleep medication use and sleep duration complaints) have been reported previously.

Given their reliance on easily understandable sleep-related variables derived from ISI and PSQI responses and frequently reported in outcome studies [63,64], our subtypes might be an aid to tailor the delivery of evidence-based treatments such as cognitive behavioral therapy for insomnia (CBT-I) [65,66,67] consistently with the stepped-care approach proposed by Rybarczyk and Mack [68]. For example, those in the SI profile might require full, multicomponent CBT-I with particular emphasis on the cognitive therapy component to manage sleep-related distress and worry, and possible psychological comorbidity should be evaluated and addressed with these patients. On the other hand, those reporting moderate insomnia symptoms with primary complaints concerning sleep duration and continuity might be treated with abbreviated CBT-I with a major focus on behavioral interventions [66] or single behavioral interventions if appropriate (e.g., sleep restriction therapy or stimulus control). Conversely, those reporting milder symptoms (SubI-SL and SubI subgroups) could benefit from less-intensive treatments or preventive programs (in individual or group format) comprising single-component interventions or structured sleep education, as already proposed for university students [69]. Finally, those in the MI-MU profile are candidates for a thorough medical assessment of their drug use to evaluate whether a supervised medication discontinuation program in association with CBT-I [70] is indicated. In addition, profile characteristics could be used to guide and optimize the assessment procedure when appropriate. Subjects reporting pronounced sleep duration and sleep continuity complaints (MI-SD profile), for example, could be evaluated with objective methods in order to ascertain whether they present insomnia with objective short sleep duration [7], whereas an extensive functional and psychometric assessment might be indicated for cases in the SI profile.

Thus, the identified and described insomnia subtypes might be informative for both research and clinical practice. Future studies should investigate their consistency and stability over time and their perspective association with a broader set of health outcomes using appropriate data-driven approaches for longitudinal research, such as latent transition analysis [71], both in community and clinical samples.

Moreover, the presented results are fundamentally exploratory and preliminary, and this study has several limitations that should be considered cautiously. Firstly, the sample was made up of university students from a single Italian university, and a remarkably higher proportion of participants were women (88% women vs. 12% men). Thus, the generalizability of the results is limited, and the proposed insomnia subtypes should be replicated on diverse samples to better understand their relevance. In addition, we did not investigate possibly relevant information that might have had an impact on the observed results, such as medical and psychiatric comorbidities, other sleep-related conditions (e.g., obstructive sleep apnea, circadian rhythm disorders, and sleep-related movement disorders), and drug and illicit substance use. Concerning the study methodology, sleep variables were solely evaluated by means of retrospective self-report questionnaires although the reliability of these instruments might be limited in comparison with other subjective (sleep diaries) or objective (polysomnography or actigraphy) measures [72,73], and the latter provide information about sleep parameters (i.e., objective duration) that have been documented to be relevant in identifying insomnia phenotypes [7]. Furthermore, latent profile analysis relied on a narrow set of variables in this study, and several sleep and non-sleep factors that may play a significant role in insomnia according to previous research have not been considered (e.g., sleep-hygiene [74], sleep-related cognitions [75], and positive and negative affect [76]). Lastly, despite the use of sleep medication representing a relevant factor in distinguishing the presented subgroups, it was evaluated with a single item from the PSQI, and possibly, relevant information about this aspect was not obtained (e.g., type of medication, duration of consumption, whether it was prescribed by a physician or not, etc.).

## 5. Conclusions

In summary, this study found five heterogeneous subtypes of insomnia presentation in a sample of university students reporting insomnia complaints. The subtypes were identified on the basis of symptoms severity and standard components of sleep quality evaluated through self-report measures, were characterized by diverse sleep complaints, and were shown to be associated with different levels of impairment and psychological difficulties. Since they can be easily identified in routine insomnia assessment, these five subtypes might help target and optimize insomnia treatment. However, further studies are needed to confirm our subtypes’ consistency and suitability for clinical practice.

## Figures and Tables

**Figure 1 jcm-11-04069-f001:**
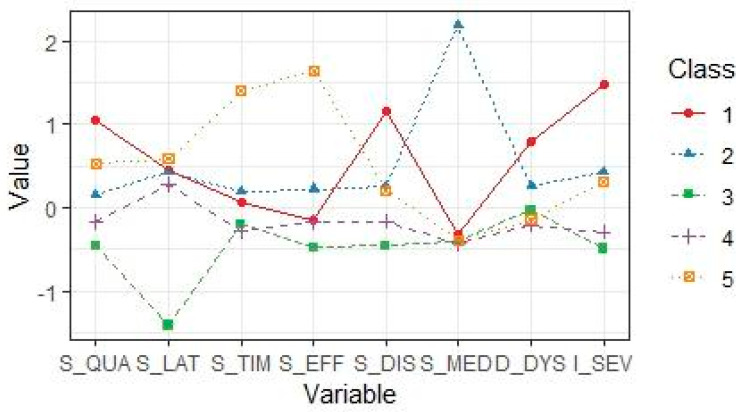
Graphical representation of the five-classes latent profile solution. Indicator variables (PSQI components and ISI total scores) labels are reported on the *x*-axis and *z*-scores on the *y*-axis. Class 1, *severe insomnia* (SI); Class 2, *moderate insomnia with medication use* (MI-MU); Class 3, *subthreshold insomnia* (SubI); Class 4, *subthreshold insomnia with sleep latency complaints* (SubI-SL); Class 5, *moderate insomnia with sleep duration complaints* (MI-SD).

**Table 1 jcm-11-04069-t001:** Participants’ (*n* = 490) socio-demographic, health-related, and sleep characteristics.

Variable	*N* (%)	Mean (*SD*)
Sex		
Women	432 (88.2)	
Men	58 (11.8)	
Age		23.4 (2.4)
Study course		
Undergraduate	411 (83.9)	
Postgraduate	79 (16.1)	
Occupational situation		
Full-time student	363 (74.1)	
Part-time job	92 (18.8)	
Full-time job	35 (7.1)	
Living situation		
With parents	354 (72.2)	
With roommates/partner	117 (23.9)	
Alone	19 (3.9)	
Tobacco use		
Yes	252 (51.4)	
No	238 (48.6)	
Excessive alcohol consumption		
≥2 per week	64 (13.1)	
<2 per week	426 (86.9)	
Physical exercise		
≥2 per week	180 (36.7)	
<2 per week	310 (63.3)	
BMI (kg/m^2^)		22.5 (4.0)
<18.5 (underweight)	60 (12.2)	
18.5 to 24.99 (normal weight)	335 (68.4)	
≥25 (overweight)	95 (19.4)	
Total sleep time (hours)		6.5 (1.1)
Sleep-onset latency (minutes)		48.5 (37.5)
Sleep efficiency index (%)		80.5 (11.9)

*Note*. BMI, body mass index. Sleep efficiency index in percentage was obtained dividing total sleep time for hours spent in bed.

**Table 2 jcm-11-04069-t002:** Participants’ mean ISI, PSQI, DASS-21, and SF-12 scores and mean differences between identified insomnia profiles.

	Total Sample (*n* = 490)	SI (*n* = 43) (1)	MI-MU (*n* = 78) (2)	SubI (*n* = 100) (3)	SubI-SL (*n* = 228) (4)	MI-SD (*n* = 41) (5)			
Variable	Mean (*SD*)	Mean (*SD*)	Mean (*SD*)	Mean (*SD*)	Mean (*SD*)	Mean (*SD*)	*F*	*η* ^2^	Significant Post Hoc
ISI	13.8 (3.3)	18.8 (3.1)	15.2 (3.8)	12.2 (2.2)	12.8 (2.3)	14.9 (3.3)	58.4 ***	0.33	1:2;1:3;1:4;1:5;2:3;2:4;3:5;4:5
PSQI total	9.5 (2.8)	11.6 (1.6)	12.7 (2.3)	6.6 (1.5)	8.6 (1.6)	12.7 (1.2)	216.5 ***	0.64	1:2;1:3;1:4;1:5;2:3;2:4;3:4;3:5;4:5
PSQI perceived sleep quality	1.9 (0.6)	2.6 (0.5)	2.0 (0.6)	1.7 (0.6)	1.8 (0.5)	2.2 (0.6)	30.2 ***	0.2	1:2;1:3;1:4;1:5;2:3;2:4;3:5;4:5
PSQI sleep latency	2.1 (0.9)	2.6 (0.7)	2.5 (0.7)	0.7 (0.5)	2.4 (0.5)	2.7 (0.6)	210.5 ***	0.64	1:3;2:3;3:4;3:5;4:5
PSQI sleep duration	1.0 (0.6)	1.1 (0.5)	1.1 (0.8)	0.9 (0.6)	0.8 (0.5)	2.1 (0.6)	44.2 ***	0.27	1:5;2:3;2:4;2:5;3:5;4:5
PSQI habitual sleep efficiency	1.0 (1.0)	0.9 (0.8)	1.2 (1.1)	0.5 (0.8)	0.8 (0.8)	2.7 (0.6)	57.3 ***	0.32	1:5;2:3;2:4;2:5;3:5;4:5
PSQI sleep disturbances	1.6 (0.6)	2.3 (0.5)	1.7 (0.6)	1.3 (0.5)	1.5 (0.5)	1.7 (0.5)	30.0 ***	0.2	1:2;1:3;1:4;1:5;2:3;2:4;3:5
PSQI sleep medication use	0.5 (1.0)	0.1 (0.4)	2.6 (0.5)	0.1 (0.2)	0.0 (0.2)	0.1 (0.3)	931.2 *** ^a^	0.89 ^a^	1:2;2:3;2:4;2:5
PSQI daytime dysfunction	1.5 (0.7)	2.1 (0.7)	1.6 (0.7)	1.4 (0.6)	1.3 (0.6)	1.3 (0.6)	13.3 ***	0.10	1:2;1:3;1:4;1:5;2:4
DASS-21 anxiety	9.7 (5.1)	13.8 (4.9)	11.2 (5.1)	8.1 (4.6)	9.0 (4.8)	10.3 (4.9)	13.9 ***	0.10	1:2;1:3;1:4;1:5;2:3;2:4
DASS-21 depression	12.3 (5.3)	16.1 (4.6)	13.2 (5.3)	11.4 (5.1)	11.6 (5.2)	13.0 (5.2)	8.5 ***	0.07	1:2;1:3;1:4
DASS-21 stress	14.9 (4.0)	18.0 (2.9)	15.9 (3.7)	14.1 (4.1)	14.2 (3.9)	15.2 (3.6)	11.2 ***	0.08	1:3;1:4;1:5;2:3;2:4
SF-12 PCS	50.3 (7.9)	45.7 (8.0)	48.6 (9.1)	51.6 (7.9)	51.3 (7.1)	49.4 (7.8)	6.6 ***	0.05	1:3;1:4
SF-12 MCS	30.2 (8.4)	26.5 (8.3)	29.4 (8.4)	31.1 (8.8)	30.6 (8.2)	30.5 (8.3)	2.8 *	0.02	1:3;1:4

*Note*. SI, *severe insomnia* subgroup; MI-MU, *moderate insomnia with medication use* subgroup; SubI, *subthreshold insomnia* subgroup; SubI-SL, *subthreshold insomnia with sleep latency complaints* subgroup; MI-SD, *moderate insomnia with sleep duration complaints* subgroup; ISI, Insomnia Severity Index; PSQI, Pittsburgh Sleep Quality Index; DASS-21, 21-item Depression Anxiety Stress Scale; SF-12 PCS, Short Form-12 Physical Component Summary; SF-12 MCS, Short Form-12 Mental Component Summary. * *p* < 0.05; *** *p* < 0.001. ^a^, The ANOVA and the post hoc tests were conducted on log-transformed sleep medication use scores. Reported *F* and *η*^2^ are relative to the log-transformed variable.

**Table 3 jcm-11-04069-t003:** Fit indices for the models examined with latent profile analysis.

Model	AIC	BIC	SABIC	Entropy	BLRT	BLRT *p*-Value
2 classes	10,820.77	10,925.63	10,846.28	0.83	345.7	0.01
3 classes	10,738.86	10,881.47	10,773.56	0.77	99.9	0.01
4 classes	10,121.82	10,302.18	10,165.7	0.81	635.04	0.01
5 classes	10,045.95	10,264.06	10,099.01	0.83	93.88	0.01
6 classes	10,048.15	10,304.01	10,110.4	0.80	15.8	0.17

*Note.* AIC, Akaike information criterion; BIC, Bayesian information criterion; SABIC, sample size-adjusted Bayesian information criterion; BLRT, bootstrap likelihood ratio test.

## Data Availability

The data that support the results of this study are available from the corresponding author upon reasonable request.

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
