# Peer review of "Sleep Quality and Insomnia Severity among Italian University Students: A Latent Profile Analysis"

_jcm, 2022, doi:10.3390/jcm11144069_

Round 1
Reviewer 1 Report
The authors used latent profile analysis to explore insomnia subtypes according to widely-used measures of symptoms severity and sleep quality among Italian university students. As a result, a total of five insomnia subtypes were identified: Severe insomnia (SI); Moderate insomnia with medication use (MI-MU); Subthreshold insomnia (SubI); Subthreshold insomnia with sleep latency complaints (SubI-SL); Moderate insomnia with sleep duration complaints (MI-SD). Overall, these findings highlight the relevance of sleep quality domains in identifying insomnia subtypes and might help optimize insomnia treatments. The findings may have some clinical meaningfulness; however, I have several concerns.
1.In Introduction, the authors used three paragraphs (Line 66-127) to list references that using latent profile analysis/latent class analysis in insomnia patients and summarize the main findings of each study. It makes me confused what the authors intended to express; Instead, I recommend the authors should illustrate and highlight the importance of the study and clinical meaning briefly and concisely.
2. The participants involved in this study were university students. So, what’s the characteristics of such population, and why you intended to perform latent profile analysis of insomnia subtype among them? I suspected you should mention this in the Introduction.
3. Conceptual coherence was evaluated and several fit indices were used to identify the optimal number of profiles in this study. Despite this is a data-driven study, clinical meaningfulness is still of importance. As mentioned in previous published study,
examination of entropy, the Vuong-Lo-Mendell-Rubin likelihood ratio, Bayesian information criterion, Akaike’s information criterion, and bivariate residuals, along with the clinical meaningfulness of the classes, were used in selection of the final model solution (reference: Postpartum Depression: Action Towards Causes and Treatment (PACT) Consortium. Heterogeneity of postpartum depression: a latent class analysis. Lancet Psychiatry. 2015 Jan;2(1):59-67. doi: 10.1016/S2215-0366(14)00055-8.) Therefore, I wonder if there is a better model that may be more meaningful in your study when considering clinical meaningfulness of the classes. Please consider.
4. The results of demographic characteristics of identified insomnia profiles was recommended to added in the study (or in the supplementary data), please see reference mentioned above.
5. The authors try to highlight the clinical meaning, e.g., subtypes might be an aid to tailor the delivery of evidence-based treatments like cognitive behavioral therapy for insomnia. However, the authors still should find more evidence to support your thinking.
6. Some minimum errors still exist: redundant space key (line 47, and line 402), and reference type (e.g., ref 17 and 18). Please check carefully.
Author Response
Thank you very much for your thorough review. We considered the points you suggested and tried to ameliorate the paper accordingly. Here are our detailed point-to-point responses to your observations (the latter highlighted in bold for distinction). All the changes made in the revised version of the manuscript were marked up using the “Track Changes” function of Microsoft Word and should be easily recognisable. The new entries in the references’ list have been highlighted in yellow in the updated version.
- In Introduction, the authors used three paragraphs (Line 66-127) to list references that using latent profile analysis/latent class analysis in insomnia patients and summarize the main findings of each study. It makes me confused what the authors intended to express; Instead, I recommend the authors should illustrate and highlight the importance of the study and clinical meaning briefly and concisely.
We do agree that the section dedicated to previous studies in the Introduction was probably too long and redundant. Although our aim was to synthesize previous findings to better delineate the context of our research, too many details were reported. In the revised manuscript, we thinned out this section highlighting the main findings of previous studies with particular regard to their clinical meaning and omitting the detailed description of previously identified typologies.
- The participants involved in this study were university students. So, what’s the characteristics of such population, and why you intended to perform latent profile analysis of insomnia subtype among them? I suspected you should mention this in the Introduction.
Thank you for this helpful observation. Indeed, this is a relevant missing point considering our sample (and the paper’s title as well). We added a short paragraph in the early section of the Introduction to briefly illustrate the relevance of sleep problems among university students and the importance of research regarding this population for clinical practice. In line with other reviewers’ suggestion, we also reported a couple of references concerning the impact of the recent pandemic on students’ sleep quality (“Interestingly, young adults [12] and university students in particular [13] report insomnia complaints more frequently in comparison with the general population, and according to several studies [14,15] the Covid-19 pandemic had a significant impact on this population’s sleep quality. Given that early diagnosis and treatment of sleep difficulties could prevent adverse outcomes in the long term [6], collecting data among university students is critical for both research and clinical practice.”)
- Conceptual coherence was evaluated and several fit indices were used to identify the optimal number of profiles in this study. Despite this is a data-driven study, clinical meaningfulness is still of importance. As mentioned in previous published study, examination of entropy, the Vuong-Lo-Mendell-Rubin likelihood ratio, Bayesian information criterion, Akaike’s information criterion, and bivariate residuals, along with the clinical meaningfulness of the classes, were used in selection of the final model solution (reference: Postpartum Depression: Action Towards Causes and Treatment (PACT) Consortium. Heterogeneity of postpartum depression: a latent class analysis. Lancet Psychiatry. 2015 Jan;2(1):59-67. doi: 10.1016/S2215-0366(14)00055-8.) Therefore, I wonder if there is a better model that may be more meaningful in your study when considering clinical meaningfulness of the classes. Please consider.
We appreciated this comment and the mentioned reference since it was an opportunity better to clarify our purposes. Indeed, in our intention conceptual coherence was meant to include clinical meaningfulness (i.e., we considered it as an umbrella label, subsuming clinical meaningfulness under conceptual and theoretical soundness). However, we understand that the two concepts might be distinguished to highlight that clinical aspects were considered in the choice of the optimal latent profile solution. Procedurally, this is exactly what we did: we examined all the models (2 to 6 classes) and the profiles’ centroids and found out that the statistically best-fitting model (the 5-classes solution) was indeed the most informative from a clinical point of view. Whereas the 6 classes model was confusing, more parsimonious models did not account for information that we considered to be relevant for insomnia management and treatment (e.g., the distinction between sleep latency and sleep duration was lost in the 4-classes model, and the group characterized by frequent sleep medication use was not identified in the 3-classes model).
In the revised version of the manuscript, we made explicit that we considered clinical meaningfulness in the selection of the optimal solution by stating it in the period you mentioned (“To identify the optimal number of profiles, conceptual coherence and clinical meaningfulness were evaluated and several fit indices were used [...]").
- The results of demographic characteristics of identified insomnia profiles was recommended to added in the study (or in the supplementary data), please see reference mentioned above.
In line with your suggestion, we added a dedicated table in appendix (Table A1, in Appendix A) reporting the main demographic and health-related characteristics (sex, age, physical exercise, body mass index, and tobacco and alcohol consumption) for the five insomnia profiles. The result of the one-way ANOVA considering age as dependent variable was also added in the text (we reported the results of chi-squared tests and ANOVAs directly in the text since they were immediately relevant).
- The authors try to highlight the clinical meaning, e.g., subtypes might be an aid to tailor the delivery of evidence-based treatments like cognitive behavioral therapy for insomnia. However, the authors still should find more evidence to support your thinking.
Although our results are exploratory and further studies are needed to prove whether our findings are clinically relevant, we definitely agree that the identification of the clinical meaning of the subtypes is critical even in this preliminary phase, and so we tried to convey our conclusions, but we possibly missed something. In the revised version, we added some passages in order to better support our considerations. We highlighted in particular that the proposed approach (tailoring CBT-I strategies according to individual characteristics) is consistent with a previously proposed stepped-care approach (Rybarczyk & Mack, 2011 added to the references list). Specific indications (which strategies to use for a particular presentation) are indeed supported by this model and by the already cited references (in particular Edinger et al., 2021).
In the last revised version, we also proposed that – besides informing and guiding the delivery of CBT-I – the identified subtypes could be useful to optimize the comprehensive assessment procedure by indicating what type of additional information should be obtained (“In addition, profile characteristics could be used to guide and optimize the assessment procedure when appropriate. Subjects reporting pronounced sleep duration and sleep continuity complaints (MI-SD profile), for example, could be evaluated with objective methods in order to ascertain whether they present insomnia with objective short sleep duration [7], whereas an extensive functional and psychometric assessment might be indicated for cases in the SI profile.”).
- Some minimum errors still exist: redundant space key (line 47, and line 402), and reference type (e.g., ref 17 and 18). Please check carefully.
Thank you for noticing. We corrected the misprints you found out and checked again the whole manuscript.
Reviewer 2 Report
Thank you for possibility for review the manuscript titled „Sleep Quality and Insomnia Severity among Italian University Students: A Latent Profile Analysis”
The article is of intrest and well written. The sample size is large, however sample size calculation is not provided. Explain how the study size was arrived at.
There are also some other methodological issues.
1) The students with anxiety disorder, panic attacks, and PTSD were not exluded,
2) Did authors exluded studens who use drugs , ie cocaine or ecstasy?
3) Sleep higiene was not assessed.
4) Sleep disorders ( ie OSA ) were not excluded.
5) The inclusion and exclusion criteria should be fully described .
6) The study’s design should be provided in the title or/and the abstract
7) The null hypothesess are not stated.
8) The introducion is too extensive, move some informations to discussion section. The influence of COVID-19 on insomia in students should be also mentioned (Fila-Witecka et al. . Sleepless in Solitude-Insomnia Symptoms Severity and Psychopathological Symptoms among University Students during the COVID-19 Pandemic in Poland. Int J Environ Res Public Health. 2022 Feb 23;19(5):2551. doi: 10.3390/ijerph19052551.)
Author Response
We thank you very much for your keen advice. We appreciated it and tried to ameliorate our work consistently with your suggestions. Here are our point-to-point responses to your comments (the latters highlighted in bold for distinction). All the changes made in the revised version of the manuscript were marked up using the “Track Changes” function of Microsoft Word and should be easily recognisable. The new entries in the references’ list have been highlighted in yellow in the updated version.
The sample size is large, however sample size calculation is not provided. Explain how the study size was arrived at.
As stated in the Methods section, the sample was drawn from a larger dataset (previously examined in Carpi, Cianfarani & Vestri, 2022) selecting participants reporting high ISI scores. Indeed, sample size was not computed a priori for this study (nor for the previous study on the larger sample). The reason for this choice is that our research was mainly exploratory and no a priori hypothesis about the relationships between variables or the results to be observed (e.g., the number or size of profiles, differences between profiles) were formulated. Thus, we did not considered conducting a power analysis (a priori or a posteriori) as a methodological need. However, future confirmatory studies investigating well-defined hypothesis (e.g., SI profile showing worse outcomes in comparison with other profiles) will definitely need to identify an adequate sample size.
1) The students with anxiety disorder, panic attacks, and PTSD were not exluded,
That is true. In fact, we did not adopt strict exclusion criteria (see the point below) and deliberately chose to investigate the characteristics of an hetherogeneous convenience sample. We did not assessed specific symptoms (criteria for panic disorder or PTSD) in our study and so it was not possible to identify participants at risk for specific disorders (although the DASS-21 measures features of anxiety, depression, and stress and cut-offs for severity have been previously proposed - see for example Brumby, Chandrasekara, McCoombe et al., 2011 -, their validity in discriminating between clinical and non-clinical populations has not been thoroughly evaluated to our knowledge, and so we chose not to rely on this instrument).
However, we do agree that the conditions you mentioned could be related with sleep outcomes and not controlling for them might be a limitation of our study (since psychiatric disorders might have an impact on sleep variables that we did not control for). We further address this point in the revised manuscript in the final part of the Discussion section dealing with limitations (“In addition, we did not investigate possibly relevant information that might have had an impact on the observed results such as medical and psychiatric comorbidities, other sleep-related conditions (e.g., obstructive sleep apnea, circadian rhythm disorders, and sleep-related movement disorders), and drug and illicit substance use.”).
2) Did authors exluded studens who use drugs , ie cocaine or ecstasy?
Drug and illicit substance use was not thoroughly assessed in our study. In the revised manuscript, we acknoledged this limitation in the added passage reported in the previous point.
3) Sleep higiene was not assessed.
That is true, and this might be a limitation since the relevance of sleep hygiene in insomnia has been widely reported (as well as the feasibility of sleep hygiene education as a treatment strategy). Indeed, we think that sleep hygiene habits could have been considered in the latent profile analysis among the indicator variables, and thus we highlighted this point in the revised manuscript (“Furthermore, latent profile analysis relied on a narrow set of variables in this study, and several sleep and non-sleep factors that may play a significant role in insomnia according to previous research have not been considered (e.g., sleep-hygiene [74], sleep-related cognitions [75], and positive and negative affect [76]).”) and added a specific reference (Carrión-Pantoja et al., 2022)
4) Sleep disorders ( ie OSA ) were not excluded.
That is true as well. In the revised manuscript, we highlighted this limitation in the added passage reported in point 2). Although the assessment of complex sleep disorders is beyond the scope of a cross-sectional study, we do agree that relevant information such as previous diagnosis or self-reported symptoms (for example, using the STOP-Bang questionnaire by Chung et al., 2008, for OSAs) could improve the quality of survey research on sleep disorders and we will definitely consider your observation in our next studies.
5) The inclusion and exclusion criteria should be fully described .
As previously mentioned, we did not consider strict inclusion/exclusion criteria. Students reporting relevant insomnia symptoms (ISI score ≥ 10) and younger than 35 years old were included and no further criteria were applied. To better clarify this issue, the sentence "No further exclusion criteria were applied" was added in the Methods (Participants and procedures subsection).
6) The study’s design should be provided in the title or/and the abstract
Thank you for this suggestion. We changed the text in the abstract to point out the cross-sectional design of the study (“Data were collected from 490 students reporting relevant insomnia symptoms through an online cross-sectional survey […]”).
7) The null hypothesess are not stated.
As we understand from your observation, you refer to the fact that the research question is not explicitly stated in terms of alternative hypotheses vs. null hypotheses.
In fact, given the essentially descriptive and exploratory purposes of our study, we did not hold explicit a priori hypotheses concerning the prospective results and so we decided to limit ourselves to report the aims of the research in a concise manner. Consistently with the data-driven approach we used, we did not formulate directional hypotheses and did not expect to find particular subgroups or specific differences between them (e.g., severe symptoms vs. subthreshold symptoms), and thus no hypotheses were stated.
8) The introducion is too extensive, move some informations to discussion section. The influence of COVID-19 on insomia in students should be also mentioned (Fila-Witecka et al. . Sleepless in Solitude-Insomnia Symptoms Severity and Psychopathological Symptoms among University Students during the COVID-19 Pandemic in Poland. Int J Environ Res Public Health. 2022 Feb 23;19(5):2551. doi: 10.3390/ijerph19052551.)
Thank you for this helpful suggestion. We considerably shortened the Introduction making the paragraphs dealing with previous studies more concise. We also added a short paragraph at the beginning to briefly illustrate the relevance of sleep problems among university students and the impact of the Covid-19 pandemic on this population citing the reference you mentioned (“Interestingly, young adults [12] and university students in particular [13] report insomnia complaints more frequently in comparison with the general population, and according to several studies [14,15] the Covid-19 pandemic had a significant impact on this population’s sleep quality. Given that early diagnosis and treatment of sleep difficulties could prevent adverse outcomes in the long term [6], collecting data among university students is critical for both research and clinical practice.”).
Round 2
Reviewer 1 Report
The authors have done a thorough job of answering the reviewers comments and I think this manuscript is now ready for publication.
Reviewer 2 Report
The manuscript has been corrected as recommended by the reviewer.
It can be published. Congratulations for Authors!